# Predictive modeling of hematoma expansion from non-contrast computed tomography in spontaneous intracerebral hemorrhage patients

Natasha Ironside[1]*, Kareem El Naamani[2], Tanvir Rizvi[3],
Mohammed Shifat El-Rabbi[4], Shinjini Kundu[5], Andrea Becceril-Gaitan[6],
Kristofor Pas[7], M Harrison Snyder[8], Ching-Jen Chen[9], Carl Langefeld[10],
Daniel Woo[11], Stephan A Mayer[12], E Sander Connolly[13], Gustavo Kunde Rohde[7,14],
VISTA-ICH, on behalf of the ERICH Investigators

[1]Department of Neurological Surgery, University of Virginia Health System, Charlottesville, United States; [2]Department of Neurological Surgery, Thomas Jefferson University, Philadelphia, United States; [3]Department of Radiology, University of Virginia Health System, Charlottesville, United States; [4]Department of Electrical and Computer Engineering, North South University, Dhaka, Bangladesh; [5]Department of Radiology, Washington University in St. Louis, St. Louis, United States; [6]Department of Neurological Surgery, University of Louisville, Louisville, United States; [7]Department of Biomedical Engineering, University of Virginia Health System, Charlottesville, United States; [8]Department of Neurological Surgery, Tufts University Medical Center, Boston, United States; [9]Department of Neurosurgery, Catholic Health Initiative, Omaha, United States; [10]Department of Biostatistics and Data Science, Wake Forest University School of Medicine, Winston-Salem, United States; [11]Department of Neurology, University of Buffalo, Buffalo, United States; [12]Department of Neurology, Westchester Medical Center, Westchester, United States; [13]Department of Neurological Surgery, Vagelos College of Physicians and Surgeons, Columbia University, New York, United States; [14]Department of Electrical and Computer Engineering, University of Virginia, Charlottesville, United States

*For correspondence:
ni8vb@uvahealth.org

Group author details:
VISTA-ICH See page 18

Competing interest: The authors declare that no competing interests exist.

## eLife Assessment

This study proposes a **valuable** and interpretable approach for predicting hematoma expansion in patients with spontaneous intracerebral hemorrhage from non-contrast computed tomography. The evidence supporting the proposed method is **solid**, including predictive performance evaluated through external validation. This quantitative approach has the potential to improve hematoma expansion prediction with better interpretability. The work will be of interest to medical biologists working on stroke and neuroimaging.

**Abstract** Hematoma expansion is a consistent predictor of poor neurological outcome and mortality after spontaneous intracerebral hemorrhage (ICH). An incomplete understanding of its biophysiology has limited early preventative intervention. Transport-based morphometry (TBM) is a mathematical modeling technique that uses a physically meaningful metric to quantify and visualize discriminating image features that are not readily perceptible to the human eye. We hypothesized

that TBM could discover relationships between hematoma morphology on initial Non-Contrast Computed Tomography (NCCT) and hematoma expansion. 170 spontaneous ICH patients enrolled in the multi-center international Virtual International Trials of Stroke Archive (VISTA-ICH) with time-series NCCT data were used for model derivation. Its performance was assessed on a test dataset of 170 patients from the Ethnic/Racial Variations of Intracerebral Hemorrhage (ERICH) study. A unique transport-based representation was produced from each presentation NCCT hematoma image to identify morphological features of expansion. The principal hematoma features identified by TBM were larger size, density heterogeneity, shape irregularity, and peripheral density distribution. These were consistent with clinician-identified features of hematoma expansion, corroborating the hypothesis that morphological characteristics of the hematoma promote future growth. Incorporating these traits into a multivariable model comprising morphological, spatial, and clinical information achieved an AUROC of 0.71 for quantifying 24 hr hematoma expansion risk in the test dataset. This outperformed existing clinician protocols and alternate machine learning methods, suggesting that TBM detected features with improved precision than by visual inspection alone. This pre-clinical study presents a quantitative and interpretable method for discovery and visualization of NCCT biomarkers of hematoma expansion in ICH patients. Because TBM has a direct physical meaning, its modeling of NCCT hematoma features can inform hypotheses for hematoma expansion mechanisms. It has potential future application as a clinical risk stratification tool.

## Introduction

Within hours of spontaneous intracerebral hemorrhage (ICH) onset, hematoma expansion contributes to mass effect and injury to the surrounding brain (*Rzepliński et al., 2022*; *Fisher, 1971*; *Mayer et al., 2021*). It is a preventable predictor of poor neurological outcome and mortality (*Rzepliński et al., 2022*; *Fisher, 1971*, *LoPresti et al., 2014*). Biophysical hypotheses for hematoma expansion are primarily derived from small pathological studies and have not been proven in the clinical setting (*Rzepliński et al., 2022*; *Fisher, 1971*). Although several non-contrast computed tomography (NCCT) features for hematoma expansion have been independently described by clinicians (i.e. swirl sign, blend sign, island sign), a quantitative method for analyzing hematoma morphology from NCCT is lacking (*Kundu et al., 2018*; *Kundu et al., 2020*; *Kolouri et al., 2017*; *Huttner et al., 2022*). Recent results demonstrating the benefit to ICH surgical evacuation motivate new approaches to enable early detection of hematoma expansion and reduce time-to-intervention in future ICH trial designs (*Naidech et al., 2022*; *Yogendrakumar et al., 2020*; *Wang et al., 2019*; *Shoamanesh et al., 2018*). Understanding the relationship between NCCT changes in hematoma morphology and the underlying expansion mechanism will be crucial to identifying preventative therapies (*Mayer et al., 2021*; *Naidech et al., 2022*; *Yogendrakumar et al., 2020*).

Qualitative NCCT markers of hematoma expansion named by clinicians have been incorporated into clinical scoring systems to predict hematoma expansion. However, their use of subjective criteria has led to scoring variability (*Fisher, 1971*; *Yogendrakumar et al., 2020*; *Kundu et al., 2020*; *Kolouri et al., 2017*; *Huttner et al., 2022*). Furthermore, use of different terminologies to describe similar features has limited our understanding of the relative diagnostic value of each feature (*Fisher, 1971*; *Yogendrakumar et al., 2020*; *Kundu et al., 2020*; *Kolouri et al., 2017*; *Huttner et al., 2022*). Deep learning methods for NCCT radiographic marker identification carry advantages of being entirely data-driven and automated (*Wang et al., 2019*). However, they are also limited by their lack of interpretability and provide little to no consideration of the biophysical processes necessary to provide a scientific rationale for their use (*Brott et al., 1997*; *Brouwers and Greenberg, 2013*). There exists a clear need for the development of a quantitative and interpretable methodology for NCCT radiographic marker identification which could improve our understanding of hematoma expansion.

Transport-based morphometry (TBM) is a quantitative modeling technique that generates a three-dimensional representation of the entire information content within an image (*Kundu et al., 2018*; *Kundu et al., 2020*). TBM subsumes well-established image features used in protocols, while also considering features not readily discernible to the human eye (*Kundu et al., 2018*). Model inversion permits visualization of discriminating morphological and spatial information (*Kolouri et al., 2017*). In this pre-clinical study of segmented time-series NCCT hematoma images, we hypothesized that transport-based morphometry (TBM) could discover relationships between NCCT morphometric

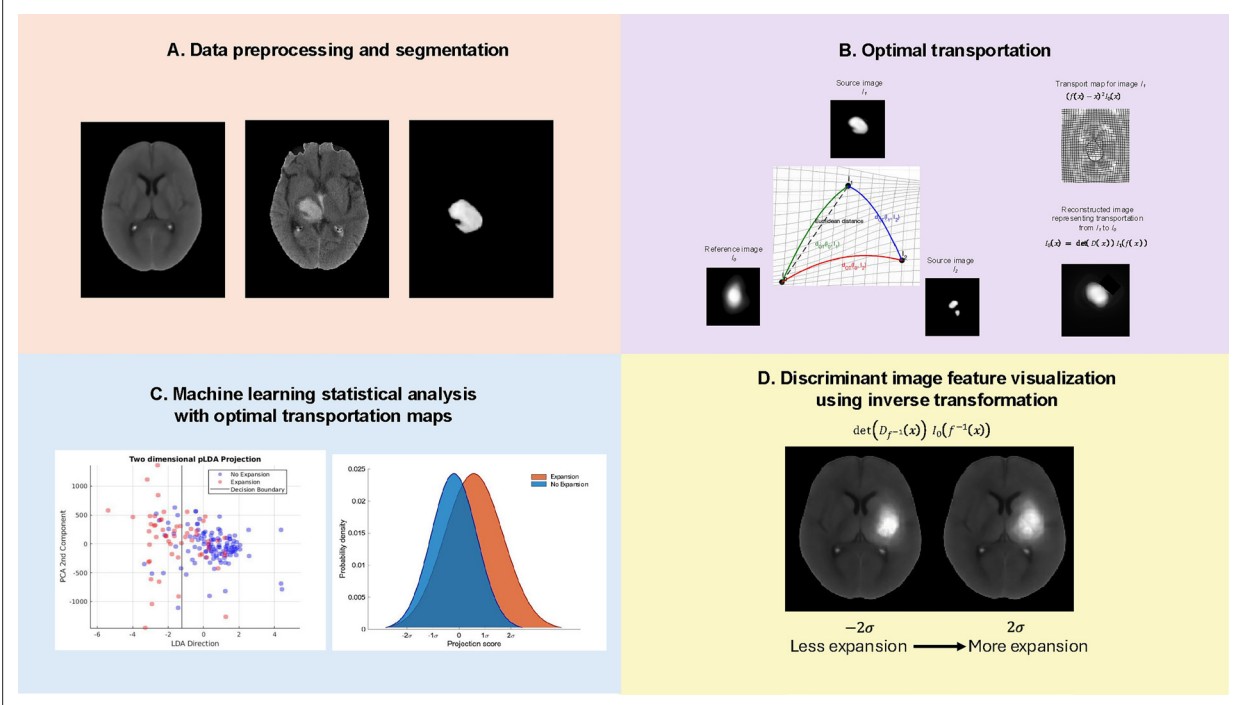

**Figure 1.** Example of the transport-based morphometry workflow. (**A**) NCCT scan registration and segmentation. A population-based high-resolution NCCT template was used for NCCT registration prior to hematoma region segmentation. (**B**) Optimal transportation. Segmented hematoma regions, depicted as source images ($I_1$, $I_2$...), are transformed to the transport domain by *pushing* mass, represented as pixel intensity, from the source image to the reference image. This process computes optimal transportation maps, thereby representing the images as points on a high-dimensional Riemannian manifold. (**C**) Machine learning statistical analysis. In transport space, differences between given source images ($I_1$, $I_2$...) can be represented as a *linear embedding* of the difference between their computed transport maps. This permits effective application of statistical analysis methods to the high-dimensional data. (**D**) Discriminant image feature visualization. Representing data as points on a Riemannian manifold enables any point to be interrogated and inverted from the transport domain to the native image domain. This generates images of the discriminant features captured during statistical analysis. TBM is performed on volumetric NCCT images and two-dimensional slices are demonstrated for illustrative purposes. Abbreviations: PCA = principal components analysis, NCCT = non-contrast computed tomography, LDA = linear discriminant analysis.

features and hematoma expansion (*Morotti et al., 2019*; *Morotti et al., 2020*; *Lv et al., 2021*; *Wang et al., 2015*). Data from the ICH section of the multicenter Virtual International Trials in Stroke Archive (VISTA-ICH) was used to derive the model, and the Ethnic/Racial Variations of Intracerebral Hemorrhage (ERICH) study was used for external validation (*Basu et al., 2014*; *Hemphill, 2019*; *Morotti et al., 2018*). We demonstrate that (1) TBM model regression can quantify changes in NCCT hematoma morphology to estimate risk of expansion, (2) TBM model inversion permits visualization of NCCT features of expansion to inform hypotheses for its biophysical mechanisms, and (3) a resulting predictive model for future expansion can outperform conventional clinician protocols and machine learning methods. An outline of the TBM workflow and proposed analytical methods is presented in *Figure 1*.

## Results
### Dataset composition
Of the 265 spontaneous ICH patients in the VISTA repository with available presentation NCCT scans, 95 were excluded (26 for corrupted/unreadable DICOM files, 3 for surgical evacuation, 9 for infra-tentorial location, 3 for no 24±6 hr interval NCCT scan, and 52 for initial ICH volume <7 mL). The remaining 170 patients (mean (SD) age 64.08 (12.45) years; 37.6% (n=64) female), comprised the derivation (training and internal validation) dataset. In the derivation dataset, the mean (SD) and median [IQR] hematoma volumes at presentation were 31.31 (24.06) and 25 [14-39] mL, respectively. The mean (SD) and median [IQR] hematoma volumes at 24±6 hr were 39.56 (34.42) and 28 [14-54] mL, respectively. Hematoma expansion was present in 32.9% (n=56) patients. Of the 3,000 spontaneous

**Table 1.** Comparison of demographic and clinical information between the expansion and no expansion groups in the derivation and test datasets.

Abbreviations: INR = international normalized ratio, S.D. = standard deviation, n=number, IVH = intraventricular hemorrhage, NCCT = non-contrast computed tomography, IQR = interquartile range. * on admission.

| | Derivation dataset (n=170) | | | Test dataset (n=170) | | |
|---|---|---|---|---|---|---|
| | Expansion (n=56) | No expansion (n=114) | p-value | Expansion (n=56) | No expansion (n=114) | p-value |
| **Demographics** | | | | | | |
| Age, years, mean ± S.D | 66.16±11.89 | 63.06±12.64 | 0.127 | 59.35±13.45 | 62.0±13.32 | 0.233 |
| Female, n (%) | 19 (33.9) | 45 (39.5) | 0.483 | 14 (25.9) | 35 (31.0) | 0.503 |
| Race/Ethnicity | | | | | | 0.837 |
| Black, n (%) | | | | 11 (20.4) | 23 (20.3) | |
| Hispanic, n (%) | | | | 22 (40.7) | 51 (45.1) | |
| White, n (%) | | | | 21 (38.9) | 39 (34.5) | |
| **Biochemistry** | | | | | | |
| INR*, mean ± S.D. | 1.07±0.26 | 1.10±0.19 | 0.564 | 1.18±0.59 | 1.08±0.29 | 0.160 |
| **Clinical parameters** | | | | | | |
| SBP*, mean mmHg ± S.D. | 185±30 | 181±28 | 0.464 | 188±37 | 186±36 | 0.723 |
| **Radiographic parameters** | | | | | | |
| Time from symptom onset to NCCT, mean min ± S.D. | 104.13±37.7 | 113.58±41.3 | 0.151 | 505.11±527.27 | 934.35±1059.37 | **0.006** |
| Hematoma growth rate, mean mL/min ± S.D. | 0.426±0.279 | 0.281±0.375 | **0.012** | 0.152±0.202 | 0.004±0.018 | **<0.001** |
| IVH score, median [IQR] | 0 [0–2] | 0 [0–1] | 0.405 | 0 [0–2] | 1 [0–2] | 0.107 |

ICH patients in the ERICH study, 1066 met inclusion criteria and were randomly sampled to generate a test (external validation) dataset of 170 patients, (mean (SD) age 61.14 (13.38) years; 29.2% (n=50) female), 20.0% (n=34) Black, 42.9% (n=73) Hispanic, and 35.3% (n=60) White. In the test dataset, the mean (SD) and median [IQR] hematoma volumes at presentation were 25.73 (19.73) and 21 [11-33] mL, respectively. The mean (SD) and median [IQR] hematoma volumes at 24±6 hr were 31.59 (24.24) and 26 [12-43] mL, respectively. Hematoma expansion was present in 32.9% (n=56) patients. A flow diagram of the patient selection process for the derivation and test datasets is shown in *Figure 2—figure supplement 1*. Comparisons of baseline demographic and clinical characteristics between the expansion and no expansion groups for each of the derivation and test datasets are presented in *Table 1*.

## Data preprocessing

The NCCT preprocessing and segmentation results are shown in *Figure 2A–I*. Comparisons of the native NCCT segmented hematoma images did not reveal a visually discernible difference between the expansion (*Figure 2J*) and no expansion groups (*Figure 2K*). The intrinsic mean template $I_{0\mu}$ used for the optimization of the linear optimal transportation framework is shown for the original and location-adjusted datasets in *Figure 2—figure supplement 2*.

## TBM model regression quantified 24-hr hematoma growth from NCCT

In the internal validation cohort of the derivation dataset, the mean correlation coefficient (CC) for the most correlated direction in transport space, $w_{corr}$, between presentation hematoma features and 24 hr absolute hematoma volume increase was 0.191 [0.184–0.198], p<0.0001 for TBM alone. This improved to 0.278 [0.271–0.285]; p<0.0001 after location and clinical information were included in the TBM model (*Figure 3—figure supplement 1*). Stepwise optimization results for the preliminary

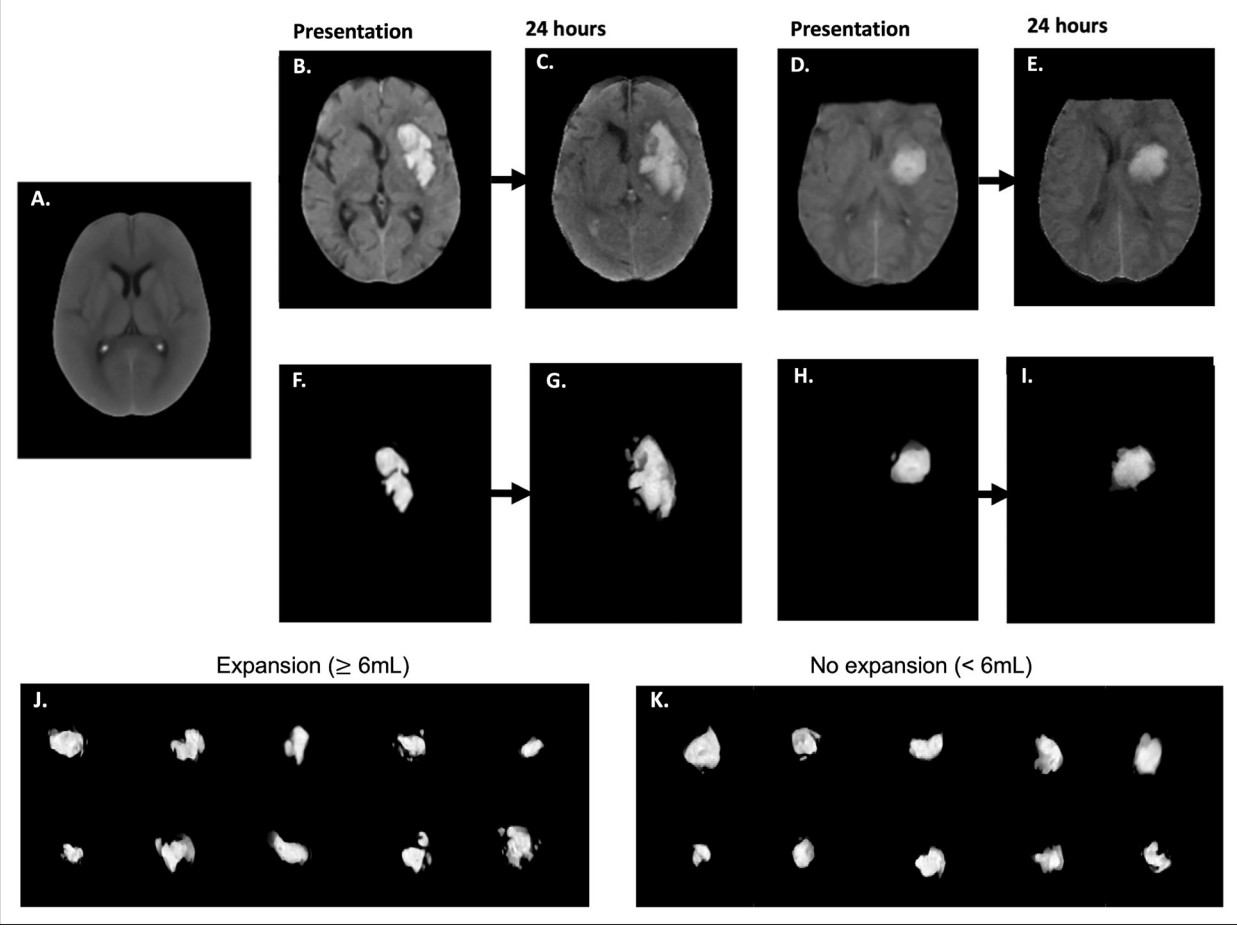

**Figure 2.** Example of the preprocessing protocol. (**A**) All NCCT scans were skull stripped and registered to a population-based high resolution NCCT template with dimensions of 256x256 x 256 and voxel spacings of 1x1 x 1 mm. (**B**) Example of a registered NCCT axial slice at presentation and (**C**) 24 hr in a patient with hematoma expansion. (**F**) The corresponding segmented and normalized hematoma image at presentation (**G**) and 24 hr for the same patient. (**D**) Example of a registered NCCT axial slice at presentation (**E**) and 24 hr in a patient without hematoma expansion. (**H**) The corresponding normalized hematoma image at presentation (**I**) and 24 hr in the same patient. Examples of segmented presentation non-contrast computed tomography (NCCT) hematoma images separated into groups of (**J**) hematoma expansion (≥6 mL hematoma volume increase at the 24 hr interval NCCT scan) and (**K**) no hematoma expansion (<6 mL hematoma volume increase at the 24 hr interval NCCT scan), demonstrating a lack of visually discernible difference between the two groups. Abbreviations: NCCT = non-contrast computed tomography.

The online version of this article includes the following figure supplement(s) for figure 2:

**Figure supplement 1.** Flow diagram illustrating the patient selection process.

**Figure supplement 2.** Two dimensional slice example illustrations of the template image.

hematoma growth prediction models are presented in *Figure 3—figure supplements 1 and 2*. When assessed in the test dataset, the final TBM model adjusted for location and clinical information quantified 24 hr volume increase from presentation hematoma features with a CC of 0.245; p=0.002 (*Figure 3A*).

## TBM predicted 24-hr hematoma expansion from NCCT

In the internal validation cohort of the derivation dataset, there was a separation of the mean probability distributions for the expansion and no expansion groups when projected onto the most discriminant direction in transport space $w^0$ (p<0.0001; *Figure 4 - figure supplement 5A-D*). The classifier trained on $w^0$ predicted expansion with a mean area under the receiver operating curve (AUROC) of 0.643 [0.640-0.648] for TBM alone. This improved to 0.698 [0.695-0.702] when location and clinical information were included in the TBM model (*Figure 4 - figure supplement 6A and D*). The mean accuracy, sensitivity, and specificity were 67.9% [67.6-68.2%], 51.0% [50.5-51.6%], and 77.6%

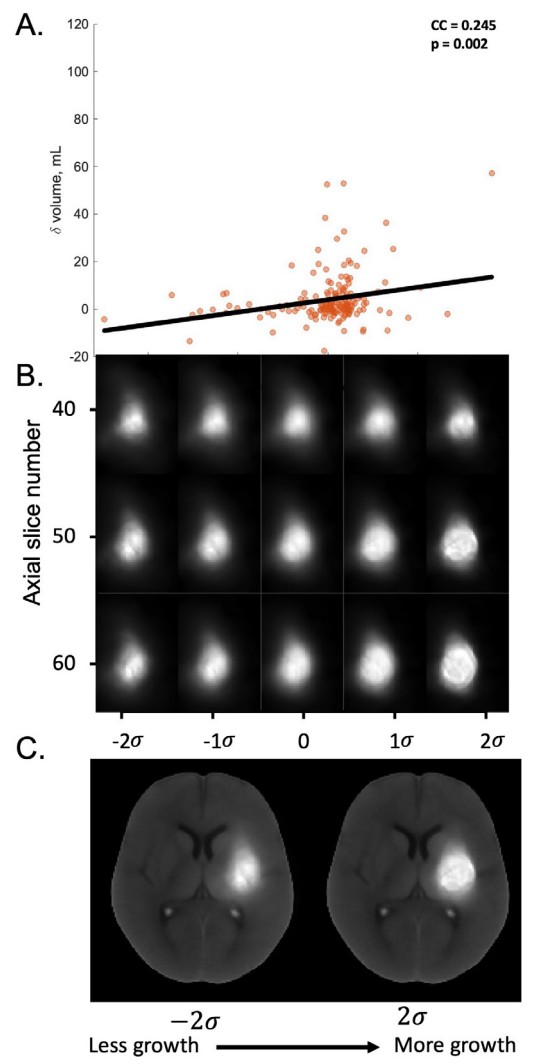

**Figure 3.** Results of the TBM model adjusted for location and clinical information in predicting 24 hr hematoma volume from the test dataset. (**A**) Scatter plots showing the relationship between the hematoma image features in the test dataset projected onto the most correlated direction $w_{corr}$ in transport space and change in hematoma volume from the presentation to the 24 hr NCCT scan. (**B**) Inverse transformations of three two-dimensional axial slice examples of the hematoma morphometric features found by the model to be associated with increasing growth, shown from left to right of the *x*-axis (**C**) Inverse transformations of the hematoma morphometric features overlaid onto the axial NCCT scan associated with *least* growth, left, and *most* growth, right. Abbreviations: NCCT = non-contrast computed tomography, TBM = transport-based morphometry, CC = correlation co-efficient $\sigma$=standard deviation of the pixel intensity distribution along $w_{corr}$.

The online version of this article includes the following figure supplement(s) for figure 3:

**Figure supplement 1.** Scatter plots showing the relationship between the hematoma image features in the internal validation cohort of the derivation dataset projected onto the most correlated direction $w_{corr}$ in transport space and change in hematoma volume from the presentation to the 24 hr NCCT scan.

**Figure supplement 2.** Scatter plots showing the relationship between the hematoma image features in the training cohort of the derivation dataset projected onto the most correlated direction $w_{corr}$ in transport space and change in hematoma volume from the presentation to the 24 hr NCCT scan.

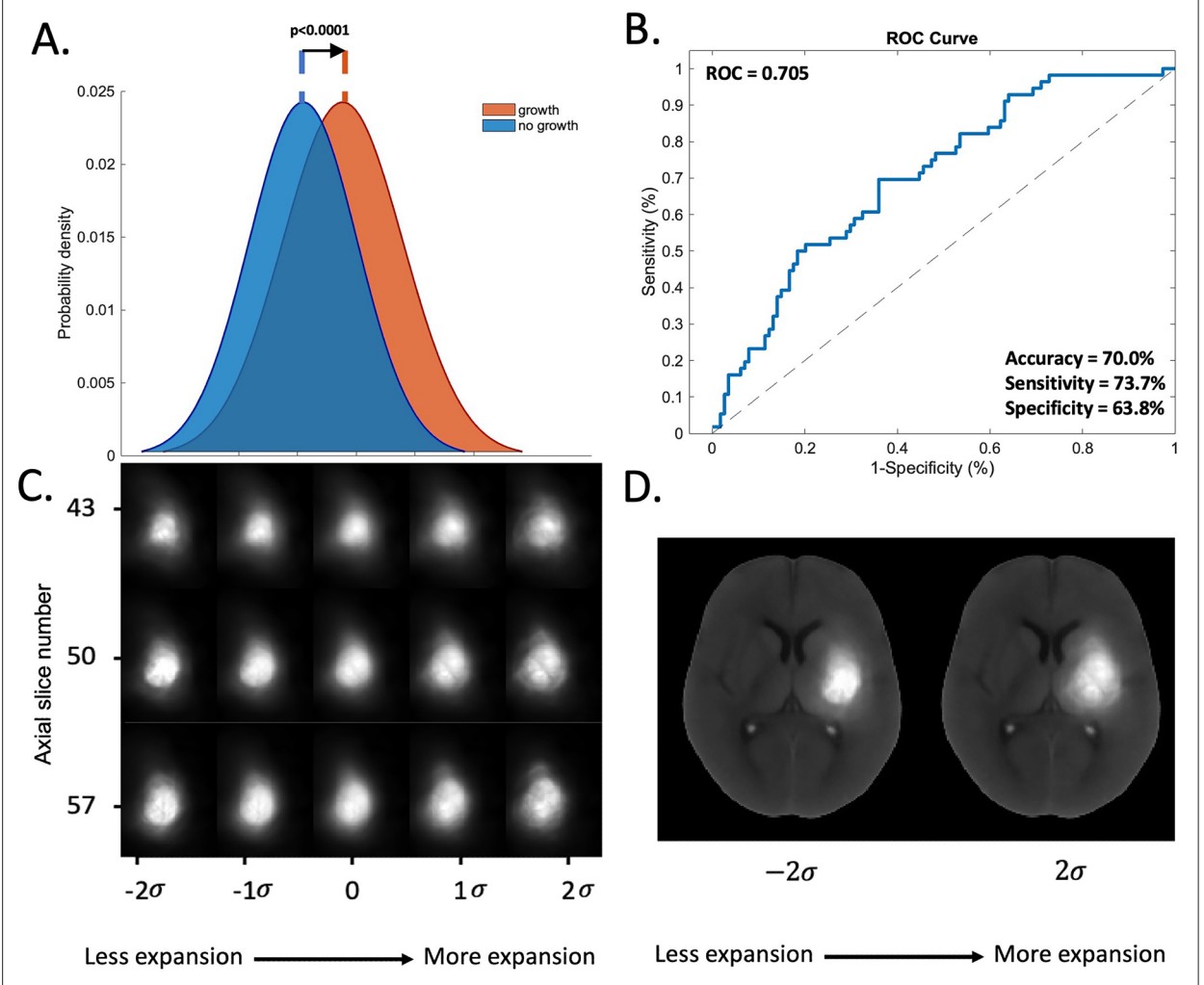

**Figure 4.** Results of the TBM model adjusted for location and clinical information in predicting 24 hr hematoma expansion from the test dataset. (**A**) Mean probability distributions of the hematoma image features in the test dataset projected onto the most discriminant direction $w_0$ in transport space showing the degree of separation between the expansion (red) and no expansion (blue) groups by the learned pLDA classifier boundary. (**B**) AUROC analyses and corresponding 95% confidence intervals of the performance of the pLDA classifier in the test dataset for (**C**) Inverse transformations of three two-dimensional axial slice examples of the hematoma morphometric features found by the model to be associated with increasing likelihood of expansion, shown from left to right of the $x$-axis (**D**) Inverse transformations of the hematoma morphometric features overlaid onto the axial NCCT scan is *least* associated with expansion, left, and *most* associated with expansion, right. Abbreviations: NCCT = non-contrast computed tomography, TBM = transport-based morphometry, AUROC = area under the receiver operator curve, pLDA = penalized linear discriminant analysis, $\sigma$=standard deviation of the pixel intensity distribution along $w_0$.

The online version of this article includes the following figure supplement(s) for figure 4:

**Figure supplement 1.** Mean probability distributions of the hematoma image features in the internal validation cohort of the derivation dataset projected onto the most discriminant direction $w_0$ in transport space showing the degree of separation between the expansion (red) and no expansion (blue) groups by the learned pLDA classifier boundary.

**Figure supplement 2.** AUROC analyses of the performance of the pLDA classifier.

**Figure supplement 3.** Mean probability distributions of the hematoma image features in the training cohort of the derivation dataset projected onto the most discriminant direction $w_0$ in transport space showing the degree of separation between the expansion (red) and no expansion (blue) groups by the learned pLDA classifier boundary.

**Figure supplement 4.** AUROC analyses of the performance of the pLDA classifier.

**Figure supplement 5.** Mean probability distributions of non-contrast computed tomography hematoma features in the internal validation cohort of the derivation dataset showing the degree of separation between the expansion (red) and no expansion (blue) groups.

**Figure supplement 6.** Mean probability distributions of non-contrast computed tomography hematoma features in the training cohort of the derivation dataset showing the degree of separation between the expansion (red) and no expansion (blue) groups.

*Figure 4 continued*

**Figure supplement 7.** AUROC analyses of the performance of each of the visually identified image features as independent predictors of hematoma expansion with comparison to the final TBM model.

[77.3-77.9%], respectively, for the clinical information and location-adjusted TBM model. Stepwise optimization results for the preliminary expansion prediction models are presented in *Figure 4— figure supplements 1–4*.

When assessed in the test dataset, the final clinical information and location-adjusted TBM model achieved a significant separation between the mean probability distributions for the expansion and no expansion groups when projected onto $w^0$ (p<0.0001; *Figure 4A*). This corresponded to an AUROC of 0.705 for discriminating expansion from no expansion (*Figure 4B*). In the test dataset, the accuracy, sensitivity, and specificity were 70.0%, 73.7%, and 63.8%, respectively.

## TBM discovers interpretable NCCT features of 24-hr hematoma expansion and growth

In contrast to the native NCCT segmented hematoma images, TBM-generated images discovered a visually discernible difference between the expansion and no expansion groups. The inverse transformations of the hematoma features projected onto $w^0$ are shown in *Figure 4C and D*, plotted in units of SD of the pixel intensity distribution along $w^0$. The visible features that discriminated expansion were larger size, elongated shape, peripheral density distribution, and density heterogeneity.

TBM-generated images also discovered a visually discernible change in NCCT features correlating with increasing 24 hr hematoma growth. The inverse transformations of the presentation hematoma features projected onto $w_{corr}$ are shown in *Figure 3B and C*, plotted in units of SD of the pixel intensity distribution along $w_{corr}$. The features associated with more growth were larger size, peripheral density distribution, and density heterogeneity.

To objectively assess the significance of the TBM-generated NCCT features of hematoma expansion, we measured each feature from the native NCCT images. Hematoma size (p<0.0001), density heterogeneity (p<0.0001), elongated shape (p<0.0001), and peripheral density distribution (p<0.0001) demonstrated separation of the mean probability distributions between the expansion and no expansion groups in the derivation dataset (*Figure 4—figure supplements 5 and 6*).

We then assessed the predictive performance of each TBM-identified NCCT hematoma feature of expansion from the native NCCT images. The AUROC for predicting 24 hr hematoma expansion was 0.577 for hematoma size, 0.578 for density heterogeneity, 0.529 for elongated shape, and 0.560 for peripheral density distribution in the test dataset. AUROC curves for predicting 24 hr hematoma expansion from the TBM-identified NCCT image features in the derivation and test datasets are presented in *Figure 4—figure supplement 7*. The clinical information and location-adjusted TBM model outperformed each image feature in predicting 24 hr hematoma expansion in both the derivation and test datasets.

## Hematoma location independently affects hematoma expansion

Because hematoma location information improved the performance of the TBM model, we assessed its independent effect on expansion. In the internal validation cohort of the derivation dataset, there was separation of the mean probability distributions of hematoma location between the expansion and no expansion groups when projected onto $w^0$ (p<0.0001; *Figure 5—figure supplement 1C*). The mean AUROC for hematoma location discriminating the expansion and no expansion groups was 0.600 [0.597–0.603] (*Figure 5—figure supplement 1A*).

TBM-generated images discovered visibly discernible hematoma locations discriminating expansion. The inverse transformations plotted in unit vectors along $w^0$ in the axial, coronal, and sagittal planes showed hematomas associated with greater likelihood of expansion to be oriented posteriorly, inferiorly, and medially towards the thalamus, posterior limb of the internal capsule, and the atrium of the lateral ventricle (*Figure 5*).

## TBM as an alternative to clinician-based hematoma expansion prediction scores

We assessed the predictive performance of established clinician-based hematoma expansion prediction scores in our datasets. The AUROC for predicting 24 hr hematoma expansion was 0.643 for BAT, 0.652 for BRAIN, 0.565 for HEAVN, 0.569 for HEP, and 0.602 for the 10-point score in the test dataset (*Figure 6*). Comparisons between the TBM model and clinician-based scores for predicting 24 hr hematoma expansion and 24 hr hematoma growth in the derivation dataset are presented in *Figure 6—figure supplements 1 and 2*. The clinical information and location-adjusted TBM model demonstrated improved performance over clinician-based scoring methods for predicting 24 hr hematoma expansion in both the derivation and test datasets.

## TBM as an alternate to machine and deep learning methods for hematoma expansion prediction

Alternate machine and deep learning methods for hematoma expansion prediction were trained using the derivation dataset, and we assessed their predictive performance in the test dataset. The AUROC for predicting 24 hr hematoma expansion was 0.511 for K-nearest neighbors, 0.501 for support vector machine, 0.535 for logistic regression, and 0.557 for 3D ResNet convolutional neural networks in the test dataset (*Figure 6—figure supplement 3*). The clinical information and location-adjusted TBM model demonstrated improved performance over alternate machine learning and deep learning methods for predicting 24 hr hematoma expansion in the test dataset.

## Discussion

Hematoma expansion is a consistent predictor of poor neurological outcome and mortality after spontaneous ICH (*Naidech et al., 2022*; *Huttner et al., 2022*; *Morotti et al., 2021*; *Brott et al., 1997*). Although early detection of hematoma expansion is paramount to implementation of preventative therapies, knowledge of its underlying biophysical processes is lacking (*Fisher, 1971*; *Brouwers and Greenberg, 2013*). This has impaired progress towards efficient and reliable methods for detecting patients at risk for expansion (*Yogendrakumar et al., 2020*). In this study, we developed a quantitative method for investigating NCCT hematoma morphology. Our TBM framework discovered hematoma morphological changes associated with expansion and was interpretable through model inversion. This novel approach discovers hypotheses for hematoma expansion pathophysiology and has the potential to improve its clinical prediction.

We considered normalized Hounsfield Unit (HU) pixel intensity values as relative measurements of blood density and applied our transport-based morphometry (TBM) technique to model the relative intensity of each pixel in a segmented NCCT hematoma image with reference to a template (*Morotti et al., 2019*; *Morotti et al., 2020*; *Lv et al., 2021*; *Wang et al., 2015*). Thus, optimal transportation-based modeling of hematoma images provided insight into inherent physical phenomena (*Kolouri et al., 2017*; *Basu et al., 2014*). When linear models were applied to the resulting transport space representations, we found a significant correlation between NCCT hematoma morphological changes and 24 hr hematoma growth. Inclusion of clinical variables and spatial location achieved further improved performance, resulting in an AUC of 0.71 for predicting hematoma expansion. The accuracy, sensitivity, and specificity were 70%, 74%, and 64%, respectively. These findings suggest that TBM could discriminate 24 hr hematoma expansion from NCCT image features. However, future validation studies should seek to identify ideal sensitivity and specificity thresholds to effectively evaluate the model's translatability to a clinical setting.

One advantage of our method is its interpretability. Feature visualization by TBM provided radiographic insight into the process of hematoma expansion (*Basu et al., 2014*). The conventional biological model for hematoma expansion is the 'avalanche effect', first observed in a pathological study from 1971 (*Fisher, 1971*). This described the process of secondary mechanical shearing of neighboring vessels at the periphery of the hematoma, resulting in successive bleeding. It was corroborated by a more recent pathological study that observed hematomas to expand in the perivascular spaces and detach branches from surrounding tissue, resulting in secondary bleeding (*Rzepliński et al., 2022*). These proposed mechanisms of hematoma expansion have not previously been examined in a large real-time patient sample. When we inverted our TBM model to generate visualizations

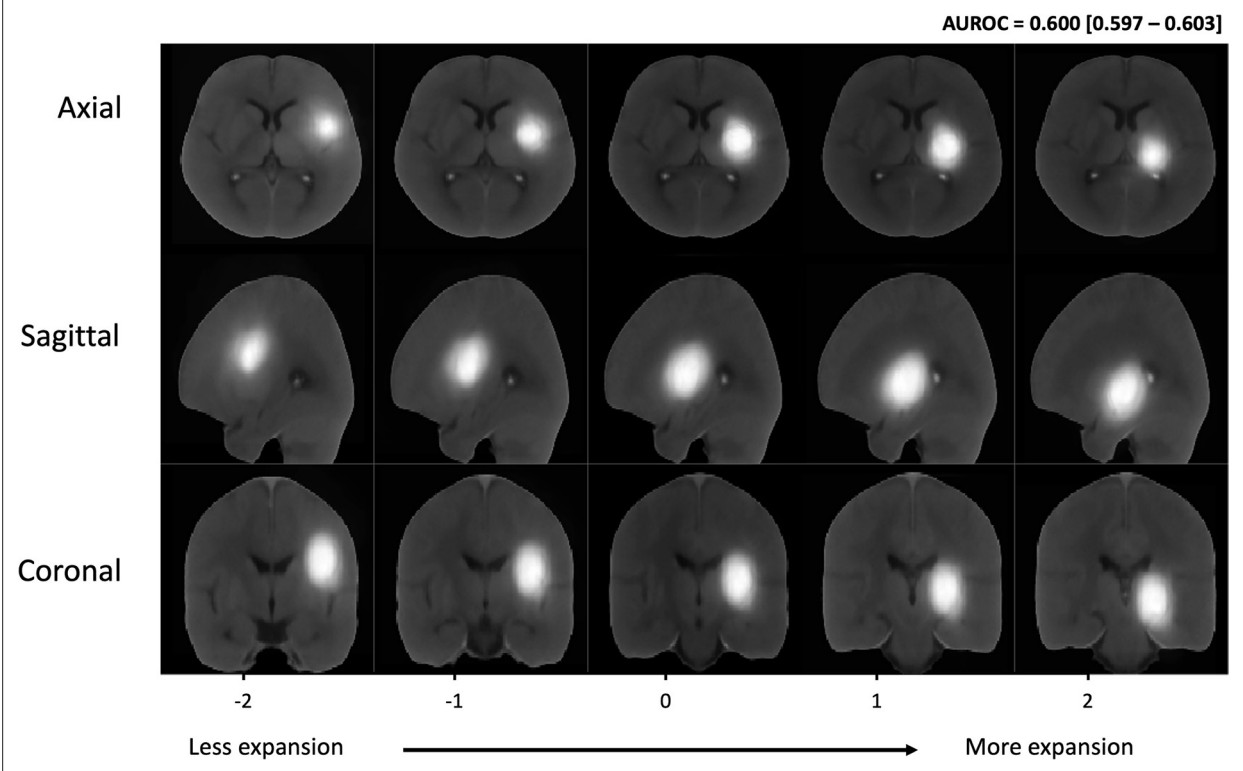

**Figure 5.** Independent effects of hematoma location as a predictor of 24 hr hematoma expansion. Two-dimensional examples of inverse transformations overlaid onto NCCT scans in the axial (top row), sagittal (second row), and coronal (third row) planes showing from left to right of the x-axis the hematoma morphometric features and location direction found by the TBM model to be associated with increasing likelihood of expansion. Abbreviations: NCCT = non-contrast computed tomography, TBM = transport-based morphometry, $\sigma$=standard deviation of the pixel intensity distribution along $w_0$.

The online version of this article includes the following figure supplement(s) for figure 5:

**Figure supplement 1.** AUROC analyses of hematoma location as an independent predictor of hematoma expansion.

of the NCCT features correlating most with expansion, we discovered morphological characteristics of expansion that were not visibly discernible on the native images. Notably, we observed preferential distribution of density towards the periphery of the expanding hematoma, which may be consistent with involvement of secondary circumferential vessels. Density heterogeneity further suggested that the hematoma expanded from bleeding at different times and locations.

While the affected location may harbor characteristic structural properties that facilitate expansion, there is conflicting evidence regarding location(s) with the propensity for expansion (*Hemphill, 2019*). Hematoma location has not yet been included as a potential modifier in expansion prediction scores (*Yogendrakumar et al., 2020*; *Brott et al., 1997*; *Morotti et al., 2018*). We defined location from the center of the hematoma, finding it to be a significant independent predictor of expansion that improved the TBM model's performance. The location most discriminating expansion was oriented towards the thalamus, posterior limb of the internal capsule, and atrium of the lateral ventricle. This correlated with the orientation of the perivascular spaces and the proposed pathological mechanism of blood product transit via these spaces, representing the path of least resistance surrounding the hematoma (*Morotti et al., 2019*). These findings lend weight to the importance of further study into the relationships between NCCT image characteristics and the biomechanics of hematoma expansion.

The shape, size, and density NCCT characteristics identified by TBM are consistent with those previously described by clinicians (*Morotti et al., 2020*; *Lv et al., 2021*). This emphasizes similarities between our TBM model and clinician-based interpretation of ICH from NCCT, corroborating their hypothesis that morphological characteristics of the hematoma promote future growth (*Yogendrakumar et al., 2020*; *Morotti et al., 2020*). In contrast to clinician-based methods, TBM overcame the subjectivity inherent to qualitative ranking, standardized the range of terminologies that

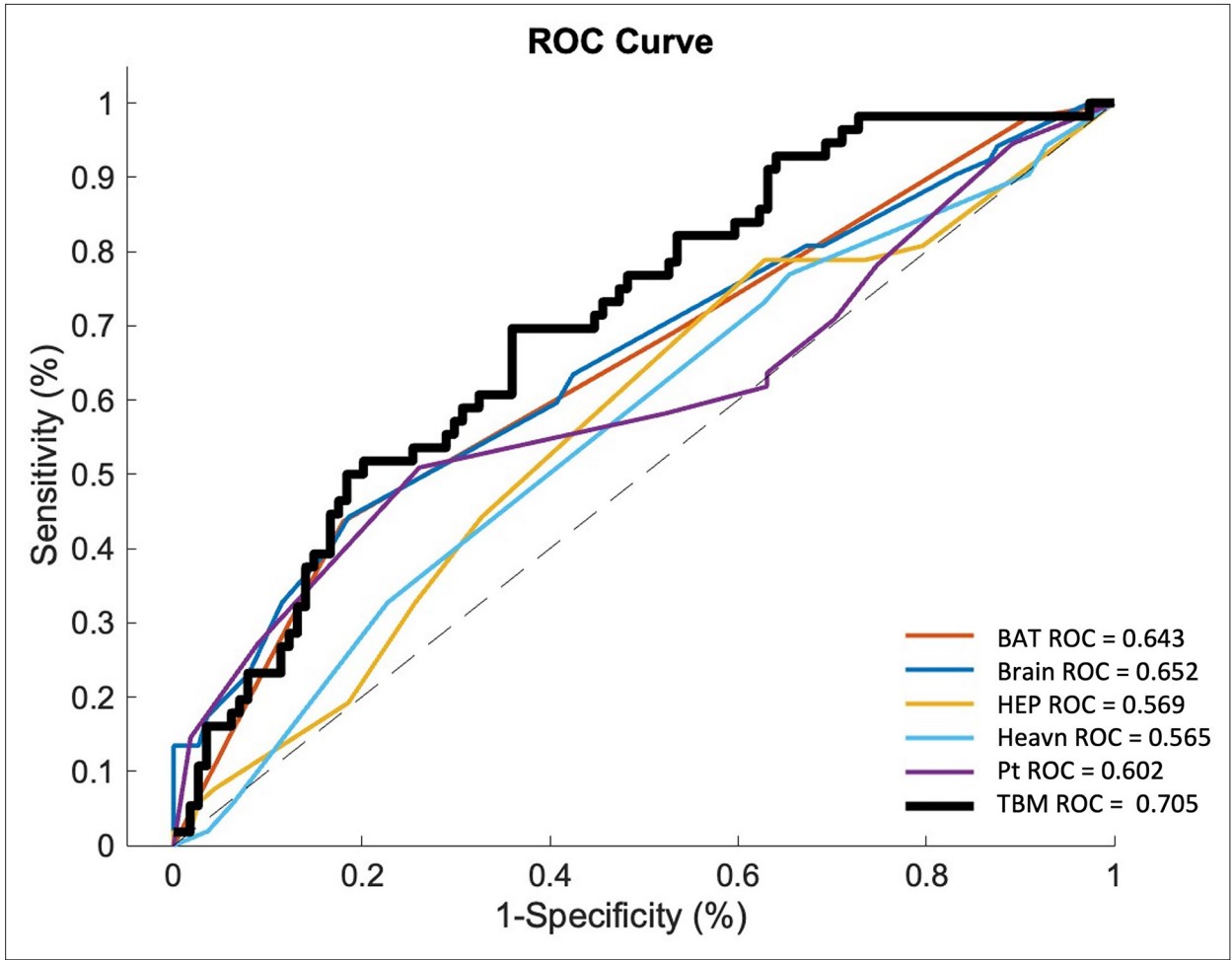

**Figure 6.** Comparisons of the performance of existing NCCT hematoma expansion prediction scores with comparison to the final TBM model adjusted for location and clinical information in the test dataset. Hematoma expansion was defined as ≥6 mL hematoma volume increase from the presentation to the 24±6 hr NCCT scan. Abbreviations: AUROC = Area Under the Receiver Operator Curve, ROC = Receiver Operator Curve, TBM = transport-based morphometry, NCCT = non-contrast computed tomography, HEAVN = Heavn score, Brain = Brain score, HEP = Hematoma expansion prediction score, Pt = 10-point score, BAT = BAT score, TBM = transport-based morphometry.

The online version of this article includes the following figure supplement(s) for figure 6:

**Figure supplement 1.** Comparisons of the performance between existing NCCT hematoma expansion prediction scores and the final TBM model adjusted for location and clinical information in the (**A**) internal validation cohort and (**B**) training cohort of the derivation dataset.

**Figure supplement 2.** Forest plots showing the mean correlation coefficient and corresponding 95% confidence intervals for each of the NCCT expansion prediction scores and 24 hr hematoma growth, measured as change in hematoma volume in milliliters from the presentation to the 24 hr NCCT scan, in the (**A**) internal validation cohort of the derivation dataset and (**B**) training cohort of the derivation dataset.

**Figure supplement 3.** Comparisons of the performance of alternate machine and deep learning methods and the final TBM model adjusted for location and clinical information in the external validation dataset.

have previously been attributed to similar image features, and permitted grading of severity. When we measured TBM-identified NCCT image features from the native image data, they were statistically significant predictors of expansion, but each was outperformed by the final TBM model in the test dataset. Similarly, TBM outperformed established clinician-based NCCT prediction scores and emerging machine learning models (*Morotti et al., 2018*; *Wang et al., 2015*; *Miyahara et al., 2018*; *Sakuta et al., 2018*; *Fu et al., 2019*). By including all information contained within a segmented hematoma image, we propose that TBM achieved greater precision and improved generalizability than pre-specified feature detection alone (*Kolouri et al., 2016*). Our findings should be interpreted with caution as the TBM method necessitates optimization of its predictive accuracy and validation in a real-world clinical setting before conclusions regarding its practical utility can be drawn. In developing

TBM, our eventual goal is to produce a fully automated hematoma expansion prediction tool. This could be used to aid clinicians in identifying patients early in the disease course who may benefit from preventative surgical and/or medical therapies. Like RAPID-AI for ischemic stroke, TBM could lead to a reliable and efficient method to select patients for timely intervention (*Vagal et al., 2019*; *Ontario Health (Quality), 2020*).

Several limitations to this study must be acknowledged. Although we included a representative multi-national population of patients enrolled in ICH clinical trials with standardized time-series NCCT data, our model derivation cohort was limited by its small sample size of 170 patients and retrospective design (*Ali et al., 2012*). Because we randomly sampled an equivalent-sized external validation dataset from the ERICH cohort study, the generalizability of the results is limited by the potential for confounding factors inherent to the validation dataset that were not accounted for. Future validation studies should investigate larger cohorts and/or matching paradigms to account for this limitation prior to drawing conclusions as to the potential clinical utility of TBM. The limited density information on NCCT and the small number of patients who experienced hematoma expansion in our dataset are likely to have affected the predictive strength of the model (*Kundu et al., 2018*; *Kundu et al., 2020*; *Miller et al., 2023*). Therefore, in spite of dimensionality reduction and external validation, our model remains at risk of overfitting. Our AUC of 0.71 in the external validation dataset, while superior to alternate clinician-based and machine learning methods, indicates moderate predictive performance and is not yet sufficient for clinical application. In addition, the poor performance of alternate machine learning methods in our test dataset warrants further investigation for potential sampling errors. The machine learning methods described may not have been adequately tuned to examine the data in the native image format with which they were presented. Future external validation studies should consider case-matched patient sampling, comparisons to optimized machine and deep learning methods, adjustment for confounders, and consideration of multiple cross-validation resampling methods prior to defining an optimal model. The relative importance of demographic and clinical predictors of hematoma expansion and the methods for incorporating these data into predictive modeling also warrant further investigation in future studies. Because our expansion definition was not based on clinical outcome, we expect future studies to investigate relationships between NCCT hematoma morphology, hematoma expansion, and neurological outcome. The derivation dataset utilized input of manually segmented images, which can be time-consuming and impractical. Future studies should continue to take advantage of fully automated hematoma segmentation methods as they undergo validation (*Ironside et al., 2022*; *Ironside et al., 2019*). As a promising preclinical study, our TBM model motivates additional external validation studies harnessing the entire ERICH dataset and prospective studies within a real-world clinical setting. These studies are in process and remain critical to define TBM's translation potential and potential impact on patient outcomes.

## Conclusions

In this pre-clinical study, we present a quantitative and interpretable approach for discovery of non-contrast computed tomography (NCCT) markers of hematoma expansion in spontaneous intracerebral hemorrhage patients. Transport-based morphometry discriminated morphological characteristics of 24 hr hematoma expansion from presentation NCCT scans. Model inversion generated visual interpretations of the features discovered by the model. This quantitative approach has the potential to improve hematoma expansion prediction. Its interpretability informs mechanisms for hematoma expansion pathophysiology.

# Materials and methods
## Study population

This analysis of spontaneous ICH patients was prepared according to the Standards for Reporting Diagnostic accuracy studies (STARD) guideline (*Ali et al., 2012*). Subjects for model derivation were recruited from the Virtual International Trials in Stroke Archive (VISTA), which is an international multi-center collaborative pooled repository of anonymized patient data from randomized clinical trials (*Ali et al., 2012*). Subjects for model external validation were recruited from the Ethnic/Racial Variations of Intracerebral Hemorrhage (ERICH) study, which is a multi-center, prospective, case-control study of ICH with emphasis on recruitment of.balanced proportions of non-Hispanic white, non-Hispanic

black, and Hispanic ICH cases (*Woo et al., 2013*). Inclusion criteria were: (1) age ≥18 years, (2) enrollment in neutral non-surgical ICH trials, (3) presentation with CT-proven ICH within 4 hr after symptom onset, (4) at least one subsequent CT scan at 24±6 hr after the initial scan, (5) available baseline clinical and laboratory data, (6) supratentorial location, and (7) initial volume ≥7 mL. Initial volume was set to distinguish microbleeds from the range of volumes associated with hematoma expansion (*Al Shahi Salman et al., 2018*). Exclusion criteria were: (1) primary intraventricular hemorrhage (IVH), and (2) ICH related to suspected secondary causes. All required Institutional Review Board reviews and approval were completed at the respective institutions.

## Clinical information

Clinical and demographic variables collected included age, sex, self-reported race/ethnicity, smoking at presentation (current or not current), previous ICH, anticoagulant use, presentation systolic blood pressure (mmHg), international normalized ratio, blood glucose (mg/dL), and time from symptom onset to first NCCT scan (min). To account for inter-patient differences in the time interval from symptom onset to presentation NCCT, hematoma growth rate was defined as: presentation hematoma volume divided by time from symptom onset to first NCCT (mL/min).

## NCCT scan preprocessing

The de-identified NCCT scans at presentation and 24±6 hr after the initial scan were transferred in Digital Imaging and Communications in Medicine (DICOM) format to a central workstation. NCCT data was pre-processed according to a standard protocol as follows: (1) 3D DICOM CT images and their corresponding segmented hematoma regions were converted to a 3-channel NumPy array; (2) windowing was performed by applying a threshold of 0–150 Hounsfield Units (HU) to the original gray-scale NCCT image; (3) the region corresponding to skull was removed using the Brain Extraction Tool from the open source FMRIB Software Library version 6.0 (Analysis Group, FMRIB, Oxford, UK); (4) each NCCT image and its corresponding segmented hematoma region were registered to a population-based, high-resolution NCCT template and re-sized to dimensions of 256x256 x 256 with voxel spacings of 1x1 x 1 mm using the Symmetric Normalization method from the Advanced Normalization Tools in Python package version 0.1.8; (5) the background of each NCCT image and its corresponding segmented hematoma region were cropped, reducing the image dimensions to 150x190 x 120; (6) segmented hematoma images located in the left hemisphere were translated across midline, so that all hematomas were registered in the right hemisphere; (7) a 3D curvature-driven Gaussian filter with a step size of 0.125 using a total of five steps was applied to smooth the segmented hematoma images; (8) each segmented hematoma image was normalized so that the sum of its intensities was equal to 1. This protocol was intended to remove differences in CT acquisition methods to permit quantitative comparisons of segmented hematoma images. All pre-processed images were visually inspected (N.I., neurosurgeon in training) to evaluate for skull-stripping and registration errors prior to further analysis.

## Hematoma segmentation

In the derivation dataset, hematoma regions were segmented by two independent raters who were blinded to outcomes information (T.R., board-certified neuroradiologist with 20 years of experience; K.E.N., neurosurgeon in training), according to our previously described manual method (*Ironside et al., 2019*). Segmentations were adjudicated by a third rater (H.S., neurosurgeon in training) and efforts were made to achieve a consensus in cases of significant inter-rater differences. In brief, the ICH hyperdensity was traced on each two-dimensional (2D) slice of the 3D DICOM image stack, using the open-source software platform 3D Slicer version 4.10.2 (National Institutes of Health, Bethesda, MD). Visual inspection, with comparison to the contralateral hemisphere, was used to differentiate ICH from IVH, subdural, and/or subarachnoid hemorrhage. In the external validation dataset, hematoma segmentation was performed according to our previously described fully automated convolutional neural network method (*Ironside et al., 2019*). In brief, the CNN architecture consisted of 31 convolutional and 7 pooling layers with a contracting and expanding topology. The rectified linear unit was used for all nonlinear functions. 50% dropout and L2 regularization were used to prevent overfitting.

## Hematoma volumetry

After NCCT scan preprocessing, ICH volumes at presentation and 24±6 hr were measured by multiplying the number of voxels (volumetric pixels) in the segmented hematoma region by the volume of each voxel (1x1 × 1 mm).

## 3D transport-based morphometry

The optimal mass transport problem seeks the most efficient way of transforming one distribution of mass to another, by minimizing a cost function (*Monge, 1781*). Transport-based morphometry (TBM) performs nonlinear image transformations by solving the continuous linear optimal transportation problem. Transforming images from their native domain to a transport domain affords three key advantages: (1) it permits discovery of discriminating image features that are not readily perceptible to the human eye, (2) it provides a physically meaningful metric, the Wasserstein distance, to quantify the relative changes in intensity between two images, and (3) it interpolates between images to generate visual interpretations of the discriminating features discovered by the model.

## Continuous linear optimal transport

We considered normalized Hounsfield Unit (HU) pixel intensity values as relative measurements of blood density and assumed that hematoma expansion occurs as a continuous process of red blood cell movement under the effect of unknown biological and physical influences (*Basu et al., 2014*). This assumption allowed us to quantify the relative movement of blood density from one image to another. Normalizing each image so that the pixel intensities sum to the same total mass allows the images to be interpreted as probability measures. In this context, mass is represented as the image intensity (*Wang et al., 2013*).

We consider two images $I_0$ and $I_1$ defined over their respective domains $\Omega_0$ and $\Omega_1$. Let $I_1$ represent an image of a segmented hematoma and $I_0$ represent the template image. Let $MP$ refer to the set of mass-preserving functions $f: \Omega_0 \to \Omega_1$ that rearranges the intensity coordinates of image $I_0$ to $I_1$. In other words: $MP := \{ f \mid det\left(D_f\left(x\right)\right) I_1 \left(f\left(x\right)\right) = I_0\left(x\right), det\left(D_f\left(x\right)\right) \geq 0 \forall x \in \Omega_0 \}$, with $D_f\left(x\right)$ representing the Jacobian matrix of deformation $f$ computed at location $I_1$ The optimal mass transport function that rearranges the intensities of the two images can be computed by solving the minimization problem:

$$\inf_{f \in MP} \int_{[0,1]^3} \left(f\left(x\right) - x\right)^2 I_0\left(x\right) dx = W_2^2\left(I_0, I_1\right) \tag{1}$$

The minimizer of the optimal transport problem stated above is known as the squared Wasserstein distance between the densities (images) $I_0$ and $I_1$, denoted above as $W_2^2\left(I_0, I_1\right)$ We utilize the previously described discretization and optimization approach to solve the optimal transport map between two digital images (*Kundu et al., 2018*; *Kolouri et al., 2016*).

The Linear Optimal Transport (LOT) framework takes $I_0$ to be a fixed reference image (*Wang et al., 2013*; *Aldroubi et al., 2021*). Given an image dataset $I_1, I_2, \cdots, I_k$, it calculates an embedding for image $I_k$ by solving for the optimal mass rearrangement problem stated above between $I_k$ and reference $I_0$. Note that given an optimal transport map $f_k$, and with knowledge of the fixed reference $I_0$, the image $I_k$ can be recovered by computing (numerically) the inverse of the function (denoted as $f^{-1}$) and computing $det\left(D_{f^{-1}}\left(x\right)\right) I_0\left(f^{-1}\left(x\right)\right) = I_1\left(y\right)$. For this reason, we can consider $f_k$ to be a new invertible representation (transform) for image $I_k$. We use the notation $\hat{I}_k = f_k$ to denote $\hat{I}_k$ the LOT transform of $I_k$. The LOT representation (feature space) is then used to perform statistical analysis using standard techniques such as principal component analysis (PCA), linear discriminant analysis (LDA), and canonical correlation analysis (CCA). Specifically, analyses performed in LOT transform space utilize the so-called LOT distance:

$$D_{LOT}^2\left(I_k, I_j\right) = \| \hat{I}_k - \hat{I}_j \|^2 = \int_{\Omega_0} \left| \hat{I}_k\left(x\right) - \hat{I}_j\left(x\right) \right|^2 dx \tag{2}$$

## Template image

The linear embedding is calculated with respect to an intrinsic mean template image $I_0$. The intrinsic mean template image $I_0$ represents the average hematoma appearance for the dataset. To estimate a reference given a set of images $I_1 \ldots I_N$, the Euclidean mean image is first computed according to:

$$\mu = 1/N \sum_{i=1}^{N} \mathrm{I_k}$$

The set of mass-preserving mappings that transform each image $\mathrm{I_k}$ into μ are computed using the minimization procedure described in *Equation 1*. The mass-preserving mappings $f_k$ for $i = 1 \ldots N$ are then averaged:

$$f(x) = 1/N \sum_{i=1}^{N} f_k(x)$$

and used to iteratively update the Euclidean mean template image $\mathrm{I_0}$, according to:

$$I_0(x) = \det(D_f(x)) \, \mu(f(x))$$

The image $\mathrm{I_0}$ is then used as the reference for the LOT calculations described above.

## Model derivation

Solving the continuous optimal transportation problem results in a unique linearly embedded 3D mass-preserving (MP) map. We obtained the MP map for the segmented presentation NCCT hematoma image of each patient. In the derivation cohort, these data were shuffled at random and separated into independent training (60%) and internal validation (40%) cohorts for model derivation and optimization. To minimize bias in data sampling, this process was repeated 1000 times to generate 1000 different training and internal validation splits. The dataset composition is described further in the Online-only supplement. Statistical analyses and data visualization were performed independently on each of the 1000 cross-validation samples, and the mean results with corresponding 95% confidence intervals are presented. The code used for model derivation is available as an open-source package (*Rohde, 2025*).

## Outcomes

Outcomes were (1) significant hematoma expansion, defined as ≥6 mL increase in hematoma volume between the presentation and 24±6 hour NCCT scans, and (2) hematoma growth, defined as absolute hematoma volume (mL) increase between the presentation and 24±6 hr NCCT scans. Absolute hematoma volume increase (mL) was chosen for its reduced susceptibility to the effect of the size of the initial hematoma and stronger association with clinical outcome than relative volume change (%). Significant hematoma expansion was set at 6 mL to reflect the threshold most used by established hematoma expansion prediction methods and in ICH clinical outcomes studies (*Mayer et al., 2021*; *Yogendrakumar et al., 2020*; *Dowlatshahi et al., 2011*). For the prediction of neurological outcome, lower hematoma expansion thresholds have been found to provide improved sensitivity at the expense of specificity, while higher thresholds provide improved specificity at the expense of sensitivity (*Mayer et al., 2021*; *Morotti et al., 2023*).

## Principal components analysis

Because the dimensionality of the hematoma image features in transport space was much higher than the number of data samples, we utilized the principal components analysis (PCA) for dimensionality reduction (*Kundu et al., 2018*; *Wang et al., 2013*). PCA is described in detail in the Online-only supplement. We retained the top *n* directions that explained 95% of the variance in the data. This eliminated the components with little contribution to the overall variance, reduced the likelihood of overfitting, and maintained separation of the training, internal validation, and test datasets.

## NCCT features discriminating hematoma expansion

To assess the relationship between presentation NCCT hematoma image features and 24 hr hematoma expansion, we used the penalized linear discriminant analysis (pLDA) method (*Kolouri et al., 2016*). Penalized Linear Discriminant Analysis (PLDA) is a modification of the Fisher linear discriminant analysis, described for morphometric studies of living tissues (*Basu et al., 2014*). This is described in detail in the Online-only supplement.

We dichotomized patients into expansion and no expansion groups by the definition of ≥6 mL and <6 mL increase in hematoma volume from the presentation to the 24 hr NCCT scan (*Mayer et al., 2021*; *Yogendrakumar et al., 2020*; *Dowlatshahi et al., 2011*). We retained the top pLDA direction $w^0$ that maximally separated the two groups to train a classifier to assign patients to the expansion or no expansion groups. The reduced dimensionality matrix of the internal validation dataset, $X_{test}$ was then projected onto $w^o$ such that $Xw_{test} \in \mathbb{R}^{w^0 \times N_{test}}$. The independent $t$-test was used to assess the degree of separation between the histogram means of the expansion and no expansion groups in $Xw_{test}$ and area under the receiver operator characteristic curve (AUROC) analyses evaluated the model's performance. Classification accuracy was assessed using sensitivity, specificity, positive predictive value (PPV), and negative predictive value (NPV).

## Relationship between NCCT features and 24-hr hematoma volume change

The canonical correlation analysis (CCA) method determined the relationship between presentation NCCT hematoma image features and 24 hr change in hematoma volume (mL; *Kundu et al., 2018*; *Wang et al., 2013*).

This is described in detail in the Online-only supplement. We retained the direction $w_{corr}$ that was most correlated with change in hematoma volume (mL) between the presentation and 24±6 hour NCCT scans. The reduced dimensionality matrix of the internal validation dataset, $X_{test}$ was then projected onto $w_{corr}$ such that $Xw_{test} \in \mathbb{R}^{w_{corr} \times N_{test}}$. The Pearson's correlation coefficient (CC) was used to assess the strength of the relationship between $Xw_{test}$ and 24 hr hematoma volume change (mL).

## Effect of location on expansion

To evaluate the independent effect of hematoma location on expansion, we defined location as the $x$, $y$, and $z$ coordinates of the center of each presentation NCCT hematoma image and used covariance matrices to represent the initial hematoma location with reference to the center of the mass of the template image $I_{0\mu}$. The above-mentioned pLDA method and its corresponding statistical analyses were used to assess the effect of location for discriminating hematoma expansion.

## Model optimization

We hypothesized that including location and clinical information would improve the model's overall performance. To test this, we rendered the MP maps location invariant by translating each presentation NCCT hematoma image according to its initial location with reference to the center of the mass of the template image $I_{0\mu}$. We compared baseline demographic and clinical information between expansion and no expansion groups using the $\chi^2$, independent $t$- or non-parametric tests, as appropriate. To generate multivariable TBM models for predicting 24 hr hematoma expansion and hematoma growth, the $x$, $y$ and $z$ co-ordinates of the translation distances (mm) and the clinical variables for $I_k$ represented as vectors $v = [v_1 \ldots v_N]^T$, were concatenated with the principal components $w_k$ of the derivation dataset $X_{train}$ and $X_{test}$, such that $a_k = [w_k | v]$. Included clinical variables were age, sex, INR at presentation, and IVH score. The resultant multivariable training and internal validation matrices are given as $aX_{train} \in \mathbb{R}^{a_k \times N_{train}}$ and $aX_{test} \in \mathbb{R}^{a_k \times N_{test}}$, respectively. Their performance for discriminating hematoma expansion and predicting future hematoma volume was assessed using abovementioned PLDA and CCA methods and their corresponding statistical analyses to obtain our final model.

## External validation

An equivalent-sized dataset of 170 patients with a hematoma expansion rate of 25–33% was estimated to have a power of 0.97–0.99 to detect an AUC of 0.7 at the 0.05 significance level, given a null AUC of 0.5. For external validation of the model, we randomly sampled eligible patients from the ERICH repository to generate a test dataset of 170 patients with an equivalent number of hematoma

expansion cases as the derivation dataset (n=56). In the test dataset, we obtained the MP map for the segmented presentation NCCT hematoma of each patient according to abovementioned methods. We performed dimensionality reduction using the abovementioned PCA method and adjusted for the relevant covariates as determined by the optimal multivariable TBM model. The trained PLDA and CCA classifiers from the derivation dataset were applied to the external validation dataset, and the optimal TBM model's performance for discriminating hematoma expansion and predicting future hematoma volume was assessed in the test dataset using abovementioned statistical analyses.

## Visualization

The continuous linear optimal transportation approach is generative, and any point in the transport space can be visualized by inverting its linear embedding. A synthetic image can be obtained from a displacement field $v$ by first computing $f_v(x) = x - v(x)$ with the following equation:

$$\det\left(D_{f_v^{-1}}(x)\right) I_0\left(f_v^{-1}(x)\right) = I_v(y)$$

where $f_v^{-1}$ denotes the inverse of the mass-preserving optimal transport map.

We inverted the projections of the derivation dataset onto the principal pLDA direction $w^0$ to visualize the presentation NCCT hematoma image features discriminating hematoma expansion at 24 hr according to:

$$w_{pLDA} = \bar{X}_{train} + \sigma w^0$$

We also inverted the projections of the derivation dataset onto the most correlated CCA direction $w_{corr}$ to visualize the presentation NCCT hematoma image features predicting greater hematoma volume increase at 24 hr according to:

$$w_{CCA} = \bar{X}_{train} + \sigma w_{corr}$$

where $w_{pLDA}$ represented the direction and magnitude by which hematoma features differed between the expansion and no expansion groups, $w_{CCA}$ represents the direction and magnitude by which hematoma image features were distributed according to changes in volume, $\bar{X}_{train}$ represents the center of $X_{train}$ and $\sigma$ is a length that has units of standard deviations of the projected data along $w^0$ or $w_{corr}$.

To visualize the effect of hematoma location on expansion, we translated the inverse transformations of the location model according to $w^o$ to visualize the $x$, $y$, and $z$ directions that discriminated hematoma expansion.

## Hematoma morphometric feature detection

We visually interpreted the inverse transformations to identify the NCCT features of hematoma expansion discovered by the 3D-TBM model. To objectively assess the significance of these TBM-generated NCCT features of hematoma expansion, they were quantified from the presentation NCCT hematoma image data in its native domain. For each subject in the derivation dataset, we measured hematoma volume, density heterogeneity, shape eccentricity, and density distribution, as described in detail in the Online-only supplement. For each NCCT feature, the independent $t$-test was used to assess the degree of separation achieved between the expansion and no expansion groups and AUROC analyses evaluated their predictive performance with comparison to the TBM model.

## Comparison to alternate NCCT clinician-based prediction scores

Established NCCT features identified by clinicians to be associated with hematoma expansion were ranked by two independent trained raters who were blinded to outcomes information (A.B-G, neurosurgeon in-training; K.E.N., neurosurgeon in-training), as described in the Online-only supplement. These features were used in combination with clinical information to compute the existing BAT, BRAIN, HEAVN, NAG, hematoma expansion prediction, and 10-point clinical hematoma expansion prediction scores in each of the derivation and test datasets (*Morotti et al., 2018*; *Wang et al., 2015*; *Miyahara et al., 2018*; *Sakuta et al., 2018*; *Fu et al., 2019*). The performance of the TBM model in predicting 24 hour expansion with comparison to each clinician-based score was assessed by AUROC analyses in the external validation dataset.

## Comparison to alternate machine and deep learning methods

We trained classical machine learning models previously described for NCCT-based classification of hematoma expansion including support vector machine, logistic regression, and k-nearest neighbors in the training dataset (*Swetz et al., 2022*; *Tanioka et al., 2022*). The de-identified NCCT scans were pre-processed and segmented according to above-mentioned methods. PCA, trained in the training dataset and projected onto the test dataset, was first used for dimensionality reduction of the segmented NCCT hematoma images. Each classifier was then trained on the dimensionality-reduced feature matrix from the training dataset. Classification performance was assessed on the dimensionality-reduced feature matrix in the test dataset. ResNet is an emerging deep learning alternative for automated prediction of hematoma expansion (*Lu et al., 2024*; *Ma et al., 2022*; *Lee et al., 2024*). We trained a 3D implementation of ResNet convolutional neural networks on the segmented NCCT hematoma images to classify hematoma expansion in the derivation dataset. The performance of the TBM model in predicting 24-hr expansion with comparison to each machine and deep learning method was assessed by AUROC analyses in the test dataset.

## Statistical analyses

Statistical analyses were performed using Stata 15.0 and Matlab R2022a. The p-values were averaged using the Fisher's method (*Fisher, 1936*). For both derivation and test datasets, statistical significance was defined as $p<0.05$.

## Acknowledgements

We wish to acknowledge Eric Oermann, MD, PhD for generously reading the draft manuscript. This research was supported by NREF Young Clinician Investigator Award and the National Institutes of Health R01GM130825-01.

## Additional information

### Group author details

**VISTA-ICH**
**Hanley DF**; **Butcher K**; **Davis S**; **Gregson B**; **Lees KR**; **Lyden P**; **Mayer S**; **Muir K**; **Steiner T**

### Funding

| Funder | Grant reference number | Author |
| --- | --- | --- |
| Neurosurgery Research and Education Foundation | Young Clinician Investigator Award | Natasha Ironside |
| National Institutes of Health | R01GM130825-01 | Gustavo Kunde Rohde |

The funders had no role in study design, data collection and interpretation, or the decision to submit the work for publication.

### Author contributions

Natasha Ironside, Conceptualization, Data curation, Formal analysis, Investigation; Kareem El Naamani, Tanvir Rizvi, Mohammed Shifat El-Rabbi, Formal analysis; Shinjini Kundu, Software, Formal analysis; Andrea Becceril-Gaitan, M Harrison Snyder, Ching-Jen Chen, VISTA-ICH, Data curation; Kristofor Pas, Carl Langefeld, Stephan A Mayer, E Sander Connolly, Writing – review and editing; Daniel Woo, Resources, Validation; Gustavo Kunde Rohde, Conceptualization, Writing – review and editing

### Author ORCIDs

Natasha Ironside  https://orcid.org/0009-0006-3130-5829
M Harrison Snyder  https://orcid.org/0000-0001-5740-2615

Reviewer #1 (Public review): https://doi.org/10.7554/eLife.105782.3.sa1

Author response https://doi.org/10.7554/eLife.105782.3.sa2

## Additional files

### Supplementary files
MDAR checklist

### Data availability
The current manuscript is a computational study, so no data have been generated for this manuscript. Data used for this manuscript is available from the ERICH and VISTA databases in accordance with the data request policies of each database. The code used for model derivation is available as an open-source package (https://github.com/rohdelab/PyTransKit/tree/master/Projects/Ironside_2025/TBM%20codes; copy archived at *Rubaiyat et al., 2025*).

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

## Appendix

### Methods

#### Dataset development

For each of the training and testing datasets, we first computed the covariance matrices $S_{train}$ and $S_{test}$, by subtracting the center of each dataset , $\bar{I}$ from $\hat{I}_k$ (**Wang et al., 2013**).

$$S = 1/N \sum_k \left( \hat{I}_k - \bar{I} \right) \left( \hat{I}_k - \bar{I} \right)^T$$

where $\hat{I}_k$ represents the MP maps for the set of hematoma images $I_k$, when $k = 1, \ldots N$. $\bar{I}$ is defined according to:

$$\bar{I} = 1/N \sum_{i=1}^{N} I_k$$

The covariance matrices can be summarized as $S_{train} \in \mathbb{R}^{p \times N_{train}}$ and $S_{test} \in \mathbb{R}^{p \times N_{test}}$ respectively, where $p$ is the number of elements in each MP map and $N_{train}$, $N_{test}$ are the total number of patients in the training and testing datasets, respectively.

#### Principal components analysis

The principal components analysis (PCA) is a method for computing the directions over which the projection of data has the largest variance in a dataset. It is given as the solution to the following optimization problem:

$$w_{PCA} = \arg \max_{|w|=1} w^T S_{train} w$$

For $k = 1 \ldots N$ in the training dataset $S_{train}$, the solution can be derived from the eigenvectors $e_k$ corresponding to the major modes of variation in the dataset and eigenvalues $\gamma_k$ corresponding to the variance of the data projected over each direction. We retained the top k PCA directions $w_k$ that preserved 95% of the variation in $S_{train}$. The reduced dimensionality matrix $X_{train} \in R^{w_k \times N_{train}}$ was then obtained by projecting the training dataset $S_{train}$ onto $w_k$. The testing dataset $S_{test}$ was also projected onto $w_k$ from the training dataset to obtain the reduced dimensionality matrix $X_{test} \in R^{w_k \times N_{test}}$.

#### Penalized Linear Discriminant Analysis

Penalized Linear Discriminant Analysis (PLDA) is a modification of the Fisher linear discriminant analysis, described for morphometric studies of living tissues (**Basu et al., 2014**). In pLDA, the directions that separate the two groups are given as the solution to the following problem:

$$w_{pLDA} = \arg\max_{|w|=1} \frac{w^T X_{train} w}{w^T \left( Xc_{train} + \alpha I \right) w}$$

where $X_{train}$ is the reduced dimensionality matrix of the training dataset and $Xc_{train}$ is the within group reduced dimensionality matrix given by:

$$Xc_{train} = \sum_c \sum_{k \in c} \left( \hat{I}_k - \bar{I}_c \right) \left( \hat{I}_k - \bar{I}_c \right)$$

where $\bar{I}_c$ represents the center of group $c$. The penalty $\alpha$ is obtained by fitting an exponential decay model to distance measurements between two consequent subspaces.

#### Canonical correlation analysis

We defined a vector $v = \left[ v_1 \ldots v_N \right]^T$ that represented the change in hematoma volume (mL) between the presentation and 24 ± 6 hr NCCT scans. In the transport space, the direction $w$ that is most correlated with $v$ is given as the solution to the following problem:

$$w_{corr} = \underset{w}{\arg\min} \frac{w^T X_{train} v}{\sqrt{w^T w}} = \frac{X_{train} v}{\sqrt{v^T X_{train}^T X_{train} v}}$$

where $X_{train}$ is the reduced dimensionality matrix of the training dataset.

## Hematoma morphometric feature detection

For $k = 1 \ldots N$ hematoma images in the total patient cohort, size was measured as the volume (mL) of the hematoma image $I_k$. Heterogeneity was measured as the entropy (bit) of $I_k$ according to:

$$I_k = -\sum_{i=0}^{n-1} p_i \log_2 p_i$$

where $n$ is the total number of intensity levels and $p_i$ is the normalized histogram count of the intensity levels in $I_k$. Shape was measured as the sphericity (%) of $I_k$ according to:

$$I_k = \frac{\pi^{\frac{1}{3}} \left(6 V_{I_k}\right)^{\frac{2}{3}}}{SA_{I_k}}$$

where $V$ is the volume (mL) and SA is the surface area ($cm^2$) of $I_k$.

Intensity distribution was measured by generating a series of four concentric contours around the center of the mass of $I_k$ according to:

$$r_n = n\left(\frac{r}{n}\right)$$

$$\left(x - cx\right)^2 + \left(y - cy\right)^2 + \left(z - cz\right)^2 \leq r_n^2$$

where $n = 4$, $r$ is the radius, $x$, $y$, $z$ are the image dimensions and cx, cy, cz are the center of the mass of $I_k$.

The mean voxel intensity value in each contour was computed to measure voxel intensity as a function of distance from the center of the mass of $I_k$ according to:

$$I_\Delta = \mu I_n - \mu I_1$$

where $I_\Delta$ is the difference between the mean intensity value of contour $n$ and contour 1.

## Results

### 3D-TBM discriminates hematoma expansion from NCCT

In the internal validation cohort of the derivation dataset, the mean area under the receiver operating curve (AUROC) for the classifier trained in transport space on the most discriminant direction $w_0$, separating the expansion and no expansion groups was 0.667 [0.663–0.671] for TBM adjusted for clinical information (*Figure 4—figure supplement 2B*), and 0.688 [0.685–0.692] for TBM adjusted for location (*Figure 4—figure supplement 2C*). The mean accuracy in the testing dataset was 0.664 [0.660–0.667] for TBM alone, 0.686 [0.683–0.693] for TBM adjusted for clinical information, and 0.662 [0.659–0.665] for TBM adjusted for location. The mean sensitivity in the internal validation cohort of the derivation dataset was 0.491 [0.485–0.496] for TBM alone, 0.525 [0.519–0.531] for TBM adjusted for clinical information, and 0.488 [0.482–0.493] for TBM adjusted for location. The mean specificity in the internal validation cohort of the derivation dataset was 0.773 [0.770–0.776] for TBM alone, 0.772 [0.769–0.775] for TBM adjusted for clinical information, and 0.761 [0.758–0.765] for TBM adjusted for location. In the internal validation cohort of the derivation dataset, there was a significant left shift in the expansion group's mean probability distribution for TBM adjusted for clinical information (p<0.0001) (*Figure 4—figure supplement 1B*), TBM adjusted for location (p<0.0001) (*Figure 4—figure supplement 1C*), and TBM adjusted for location and clinical information (p<0.0001) (*Figure 4—figure supplement 1D*), when compared to the no expansion group.

The mean AUROC in the training cohort of the derivation dataset for the classifier trained on $w_0$ was 0.780 [0.778–0.782] for TBM alone (*Figure 4—figure supplement 4A*), 0.804 [0.802–0.806] for TBM adjusted for clinical information (*Figure 4—figure supplement 4B*), 0.745 [0.743–0.747] for TBM adjusted for location only (*Figure 4—figure supplement 4C*), and 0.777 [0.774–0.779] for TBM adjusted for location and clinical information (*Figure 4—figure supplement 4D*). The mean accuracy in the training cohort of the derivation dataset was 0.741 [0.739–0.743] for TBM alone, 0.763 [0.761–0.766] for TBM adjusted for clinical information, 0.701 [0.699–0.703] for TBM adjusted for location only and 0.707 [0.705–0.710] for TBM adjusted for location and clinical information. The mean sensitivity in the training cohort of the derivation dataset was 0.598 [0.594–0.602] for TBM alone, 0.645 [0.640–0.650] for TBM adjusted for clinical information, 0.543 [0.539–0.546] for TBM adjusted for location only, and 0.553 [0.549–0.557] for TBM adjusted for location and clinical information. The mean specificity in the training cohort of the derivation dataset was 0.834 [0.832–0.836] for TBM alone, 0.829 [0.827–0.831] for TBM adjusted for clinical information, 0.842 [0.840–0.845], 0.794 [0.792–0.796] for TBM adjusted for location only, and 0.796 [0.793–0.799] for TBM adjusted for location and clinical information. In the training cohort of the derivation dataset, there was a significant left shift in the mean probability distribution of the expansion group for TBM alone (p<0.0001) (*Figure 4—figure supplement 3A*), TBM adjusted for clinical information (p<0.0001) (*Figure 4—figure supplement 3B*), TBM adjusted for location only (p<0.0001) (*Figure 4—figure supplement 3C*), and TBM adjusted for location and clinical information (*Figure 4—figure supplement 3D*), when compared to the no expansion group.

## 3D-TBM estimates future hematoma growth from NCCT

In the internal validation cohort of the derivation dataset, the mean correlation coefficient (CC) for the most correlated direction in transport space, $w_{corr}$, between the presentation hematoma features and 24 hr hematoma growth was 0.192 [0.185–0.199], p<0.0001 for TBM adjusted for clinical information (*Figure 3—figure supplement 1B*), and 0.278 [0.271–0.284], p<0.0001 for TBM adjusted for location only (*Figure 3—figure supplement 1E*).

The mean correlation coefficient (CC) in the training cohort of the derivation dataset for the most correlated direction in transport space, $w_{corr}$, between the presentation hematoma features and 24 hr hematoma growth was 0.317 [0.314–0.321], p<0.0001 for TBM alone (*Figure 3—figure supplement 2A*), 0.318 [0.314–0.321], p<0.0001 for TBM adjusted for clinical information (*Figure 3—figure supplement 2B*), 0.283 [0.279–0.288], p<0.0001 for TBM adjusted for location (*Figure 3—figure supplement 2E*), and 0.269 [0.266–0.272]; p<0.0001 for TBM adjusted for location and clinical information (*Figure 3—figure supplement 2F*).

## The independent effect of hematoma location on hematoma expansion

In the training cohort of the derivation dataset, there was a significant right shift in the mean probability distribution of the hematoma location projected onto the most discriminant direction in transport space for the expansion group, when compared to the no expansion group, (p<0.0001) (*Figure 5—figure supplement 1D*). The mean AUROC for the most discriminant hematoma location that separated the expansion and no expansion groups was 0.636 [0.634–0.639] (*Figure 5—figure supplement 1B*). The mean accuracy, sensitivity, and specificity in the training cohort of the derivation dataset were 0.576 [0.574–0.579], 0.402 [0.399–0.405] and 0.740 [0.737–0.743]. The mean accuracy, sensitivity, and specificity in the internal validation cohort of the derivation dataset were 0.557 [0.555–0.560], 0.382 [0.378–0.387] and 0.719 [0.715–0.722], respectively.

## Comparison to clinician-based scores

The mean AUROC in predicting 24 hr hematoma expansion was 0.660 [0.656–0.663] for BAT, 0.623 [0.620–0.627] for BRAIN, 0.663 [0.660–0.666] for HEAVN, 0.578 [0.575–0.581] for NAG, and 0.563 [0.561–0.567] for 10-point scores in the internal validation cohort of the derivation dataset. The mean accuracy in predicting 24 hr hematoma expansion was 63.2% [62.8–63.6%] for BAT, 59.1% [58.5–59.7%] for BRAIN, 62.9% [62.5–63.3%] for HEAVN, 51.1% [50.4–51.7%] for NAG, and 63.2% [62.8–63.7%] for 10-point scores in the internal validation cohort of the derivation dataset. The mean AUROC in predicting hematoma expansion at 24 hr was 0.664 [0.662–0.666] for the BAT, 0.628 [0.623–0.630] for the BRAIN, 0.661 [0.658–0.662] for the HEAVN, 57.7% [0.574–0.579] for the NAG,

and 0.556 [0.554–0.558] for the 10-point scores, in the training cohort of the derivation dataset (*Figure 5B*). The mean sensitivity for predicting hematoma expansion at 24 hr was 79.9% [79.7–80.2%] for the BAT, 81.2% [80.7–81.7%] for the BRAIN, 81.5% [81.2–81.7%] for the HEAVN, 82.2% [81.8–82.7%] for the NAG, and 73.9% [73.6–74.2%] for the 10-point scores, in the training cohort of the derivation dataset.

The mean CC for higher score and 24 hr hematoma growth was 0.220 [0.215–0.224] for BAT, 0.109 [0.104–0.114] for BRAIN, 0.207 [0.201–0.212] for HEAVN, 0.143 [0.137–0.148] for NAG, and 0.096 [0.089–0.102] for 10-point score, in the internal validation cohort of the derivation dataset. The mean CC for higher score and 24 hour hematoma growth was 0.226 [0.222–0.229] for the BAT, 0.109 [0.104–0.114] for the BRAIN, 0.207 [0.204–0.211] for the HEAVN, 0.145 [0.141–0.148] for the NAG, and 0.103 [0.099–0.107] for the 10-point scores, in the training cohort of the derivation dataset (*Figure 6—figure supplement 1*).

