## [Editor Report · eLife Assessment]

This study proposes a **valuable** and interpretable approach for predicting hematoma expansion in patients with spontaneous intracerebral hemorrhage from non-contrast computed tomography. The evidence supporting the proposed method is **solid**, including predictive performance evaluated through external validation. This quantitative approach has the potential to improve hematoma expansion prediction with better interpretability. The work will be of interest to medical biologists working on stroke and neuroimaging.

---

## [Referee Report · Reviewer #1 (Public review)]

Summary:

The study explores the use of Transport-based morphometry (TBM) to predict hematoma expansion and growth 24 hours post-event, leveraging Non-Contrast Computed Tomography (NCCT) scans combined with clinical and location-based information. The research holds significant clinical potential, as it could enable early intervention for patients at high risk of hematoma expansion, thereby improving outcomes. The study is well-structured, with detailed methodological descriptions and a clear presentation of results. However, the practical utility of the predictive tool requires further validation, as the current findings are based on retrospective data. Additionally, the impact of this tool on clinical decision-making and patient outcomes needs to be further investigated.

Strengths

(1) Clinical Relevance: The study addresses a critical need in clinical practice by providing a tool that could enhance diagnostic accuracy and guide early interventions, potentially improving patient outcomes.

(2) Feature Visualization: The visualization and interpretation of features associated with hematoma expansion risk are highly valuable for clinicians, aiding in the understanding of model-derived insights and facilitating clinical application.

(3) Methodological Rigor: The study provides a thorough description of methods, results, and discussions, ensuring transparency and reproducibility.

Comments on revisions:

The authors have addressed my concerns.

---

## [Author Response]

The following is the authors’ response to the original reviews.

**Reviewer #1 (Public review):**
Summary:The study explores the use of Transport-based morphometry (TBM) to predict hematoma expansion and growth 24 hours post-event, leveraging Non-Contrast Computed Tomography (NCCT) scans combined with clinical and location-based information. The research holds significant clinical potential, as it could enable early intervention for patients at high risk of hematoma expansion, thereby improving outcomes. The study is well-structured, with detailed methodological descriptions and a clear presentation of results. However, the practical utility of the predictive tool requires further validation, as the current findings are based on retrospective data. Additionally, the impact of this tool on clinical decision-making and patient outcomes needs to be further investigated.Strengths:(1) Clinical Relevance: The study addresses a critical need in clinical practice by providing a tool that could enhance diagnostic accuracy and guide early interventions, potentially improving patient outcomes.(2) Feature Visualization: The visualization and interpretation of features associated with hematoma expansion risk are highly valuable for clinicians, aiding in the understanding of model-derived insights and facilitating clinical application.(3) Methodological Rigor: The study provides a thorough description of methods, results, and discussions, ensuring transparency and reproducibility.Weaknesses:(1) The limited sample size in this study raises concerns about potential model overfitting. While the reported AUCROC of 0.71 may be acceptable for clinical use, the robustness of the model could be further enhanced by employing techniques such as k-fold crossvalidation. This approach, which aggregates predictive results across multiple folds, mimics the consensus of diagnoses from multiple clinicians and could improve the model's reliability for clinical application. Additionally, in clinical practice, the utility of the model may depend on specific conditions, such as achieving high specificity to identify patients at risk of hematoma expansion, thereby enabling timely interventions. Consequently, while AUC is a commonly used metric, it may not fully capture the model's clinical applicability. The authors should consider discussing alternative performance metrics, such as specificity and sensitivity, which are more aligned with clinical needs. Furthermore, evaluating the model's performance in real-world clinical scenarios would provide valuable insights into its practical utility and potential impact on patient outcomes.

We thank the reviewer for these thoughtful comments. We agree that k-fold cross validation is a valid approach to reduce bias associated with overfitting and account for variability in the dataset composition. During the training and optimization process, this was employed within the VISTA dataset where data were shuffled at random and separated into independent training (60%) and internal validation (40%) datasets. This process was repeated 1000 times, to generate 1000 different training and internal validation splits. Statistical analyses and data visualization were performed independently on each of the 1000 cross-validation samples, and the mean results with corresponding 95% confidence intervals are presented. The p-values were averaged using the Fisher’s method. We have included this information in the methods section. [Page 22; Paragraph 1, Lines 8-10]. External validation was performed on the ERICH dataset and analyzed only once. We chose not to perform k-fold cross validation with the test dataset in attempt to assess the model’s generalizability to unseen data from a different patient cohort. However, we agree that taking advantage of the full 1,066 ERICH cases for model validation would improve the strength of our conclusions regarding the model’s robustness. This has been included in the discussion. [Page 15; Paragraph 1; Lines 11-14].

We agree that the AUC alone will not effectively describe the clinical applicability of the intended model. We have added the sensitivity and specificity metrics for the TBM’s performance in the external dataset to the discussion. The design of the present study was primarily a pre-clinical methodological study. However, we have suggested that future external validation studies should seek to identify ideal sensitivity and specificity thresholds when evaluating the model’s translatability to a clinical setting. [Page 11; Paragraph 2; Line 22 and Page 12; Paragraph 1; Lines 2-4]. We agree that future validation studies should also assess the model’s performance in a real-world clinical setting and have emphasized this point in the discussion. [Page 13; Paragraph 2; Lines 22-23 and Page 14; Paragraph 1; Lines 1-4].

(2) The authors compared the performance of TBM with clinical and location-based information, as well as other machine learning methods. While this comparison highlights the relative strengths of TBM, the study would benefit from providing concrete evidence on how this tool could enhance clinicians' ability to assess hematoma expansion in practice. For instance, it remains unclear whether integrating the model's output with a clinician's own assessment would lead to improved diagnostic accuracy or decisionmaking. Investigating this aspect-such as through studies evaluating the combined performance of clinician judgment and model predictions-could significantly enhance the tool's practical value.

We thank the reviewer for this suggestion. The present study intended to suggest potential advantages of the TBM method with comparison to alternate clinician-based and machine learning methods. While we agree that the TBM method warrants further evaluation in a realworld clinical setting to determine its practical utility, we propose that further optimization of TBM is first needed to improve its predictive accuracy.

In developing TBM, our eventual goal is to produce a prediction tool, which can identify patients at risk for hematoma expansion early in the disease course, who may benefit from intervention with surgical and/or medical therapies. Current clinician-based risk stratification methods are highly variable in accuracy, inefficient, and require subjective interpretation of the NCCT scan. Our eventual goal is to aid clinical decision making with an automated, accurate and efficient model. In follow up work, we will study how to combine information derived from imaging and TBM with other assessment tools and clinical data in order to best inform clinicians. This has been incorporated into the discussion. [Page 14; Paragraph 1; Lines 1-4].

**Reviewer #2 (Public review):**
Summary:The author presents a transport-based morphometry (TBM) approach for the discovery of noncontrast computed tomography (NCCT) markers of hematoma expansion risk in spontaneous intracerebral hemorrhage (ICH) patients. The findings demonstrate that TBM can quantify hematoma morphological features and outperforms existing clinical scoring systems in predicting 24-hour hematoma expansion. In addition, the inversion model can visualize features, which makes it interpretable. In conclusion, this research has clinical potential for ICH risk stratification, improving the precision of early interventions.Strengths:TBM quantifies hematoma morphological changes using the Wasserstein distance, which has a well-defined physical meaning. It identifies features that are difficult to detect through conventional visual inspection (such as peripheral density distribution and density heterogeneity), which provides evidence supporting the "avalanche effect" hypothesis in hematoma expansion pathophysiology.Weaknesses:(1) As a methodology-focused study, the description of the methods section somewhat lacks depth and focus, which may make it challenging for readers to fully grasp the overall structure and workflow of the approach. For instance, the manuscript lacks a systematic overview of the entire process, from NCCT image input to the final prediction output. A potential improvement would be to include a workflow figure at the beginning of the manuscript, summarizing the proposed method and subsequent analytical procedures. This would help readers better understand the mechanism of the model.

We thank the reviewer for this suggestion. We have included a figure detailing the TBM workflow to improve reader understanding. [Figure 1, Page 5; Paragraph 2; Lines 19-20 and Page 30; Paragraph 1].

(2) The description of the comparison algorithms could be more detailed. Since TBM directly utilizes NCCT images as input for prediction, while SVM and K-means are not inherently designed to process raw imaging data, it would be beneficial to clarify which specific features or input data were used for these comparison models. This would better highlight the effectiveness and advantages of the TBM method.

We thank the reviewer for this suggestion. We have included additional details of the comparison with machine learning models in the methods section. While we used PCA on the extracted transport maps and raw image data for dimensionality reduction prior to classification, we agree that the machine learning methods described may not have been optimally tuned to examine the data in the format in which it was presented. Future studies should aim to compare TBM with optimized machine and deep learning methods to determine TBM’s potential as an automated clinical risk stratification tool. We have added this to the limitations section of the discussion. [Page 14; Paragraph 2; Lines 22-23 and Page 15; Paragraph 1; Lines 1-2].

(3) The relatively small training and testing dataset may limit the model's performance and generalizability. Notably, while the study mentions that 1,066 patients from the ERICH dataset met the inclusion criteria, only 170 were randomly selected for the test set. Leveraging the full 1,066 ERICH cases for model training and internal validation might potentially enhance the model's robustness and performance.

We thank the reviewer for this suggestion. As the reviewer highlights, the intention of the manuscript was to present a methodologically focused study which led to our small validation cohort of 170 patients from the ERICH dataset. It is our intention to further optimize and validate the TBM method in a future larger study which is underway, taking full advantage of the ERICH dataset. This has been incorporated into the discussion section. [Page 15; Paragraph 1; Lines 1114].

(4) Some minor textual issues need to be checked and corrected, such as line 16 in the abstract "Incorporating these traits into a v achieved an AUROC of 0.71 ...".

We thank the reviewer for this comment. The typographical error has been corrected.

(5) Some figures need to be reformatted (e.g., the x-axis in Figure 2 a is blocked).

We thank the reviewer for this comment. This was intentional to demonstrate that the X-axis in Figure 2a and 2b are identical and thereby highlight image features corresponding to the regression line on the graph.